# Effect of Combined Antiretroviral Therapy on the Levels of Selected Parameters Reflecting Metabolic and Inflammatory Disturbances in HIV-Infected Patients

**DOI:** 10.3390/jcm11061713

**Published:** 2022-03-19

**Authors:** Karolina Jurkowska, Beata Szymańska, Brygida Knysz, Agnieszka Piwowar

**Affiliations:** 1Department of Toxicology, Faculty of Pharmacy, Wroclaw Medical University, 50-556 Wroclaw, Poland; karolina.jurkowska@student.umw.edu.pl (K.J.); agnieszka.piwowar@umw.edu.pl (A.P.); 2Department of Infectious Diseases, Liver Diseases and Acquired Immune Deficiencies, Faculty of Medicine, Wroclaw Medical University, 50-368 Wroclaw, Poland; brygida.knysz@umw.edu.pl

**Keywords:** HIV, combined antiretroviral therapy, panel parameters, carbohydrate, lipid metabolism, cardiovascular diseases, inflammation

## Abstract

Subjects infected with human immunodeficiency virus (HIV) treated with combined antiretroviral therapy (cART) show a greater predisposition to metabolic disturbances compared to the general population. The aim of the study was to assess the effect of cART on the level of selected parameters related to carbohydrate and lipid metabolism, cardiovascular diseases and inflammation in the plasma of HIV-infected patients against the uninfected. The levels of irisin (IRS), myostatin (MSTN), peptide YY (PYY), glucagon-like peptide-1 (GLP-1), dipeptidyl peptidase IV (DPP-4), fetuin A (FETU-A), pentraxin 3 (PTX 3), chemokine stromal cell-derived factor 1 (SDF-1), and regulated on activation normal T cell expressed and secreted (RANTES) in the plasma of HIV-infected patients and the control group were measured by immunoassay methods. HIV-infected patients were analyzed in terms of CD4+ T cells and CD8+ T cell count, HIV RNA viral load, and the type of therapeutic regimen containing either protease inhibitors (PIs) or integrase transfer inhibitors (INSTIs). The analysis of HIV-infected patients before and after cART against the control group showed statistically significant differences for the following parameters: IRS (*p* = 0.02), MSTN (*p* = 0.03), PYY (*p* = 0.03), GLP-1 (*p* = 0.03), PTX3 (*p* = 0.03), and RANTES (*p* = 0.02), but no significant differences were found for DPP-4, FETU-A, and SDF-1. Comparing the two applied therapeutic regimens, higher levels of all tested parameters were shown in HIV-infected patients treated with INSTIs compared to HIV-infected patients treated with PIs, but the differences were not statistically significant. The obtained results indicated significant changes in the expression of selected parameters in the course of HIV infection and cART. There is need for further research on the clinical usefulness of the selected parameters and for new information on the pathogenesis of HIV-related comorbidities to be provided. The obtained data may allow for better monitoring of the course of HIV infection and optimization of therapy in order to prevent the development of comorbidities as a result of long-term use of cART.

## 1. Introduction

About 37.7 million people worldwide are currently infected with the human immunodeficiency virus (HIV) [1]. The introduction of combined antiretroviral therapy (cART) has proved to be a breakthrough in the treatment of HIV infection [2]. This therapy involves the use of at least three drugs from different available pharmacological groups, ensuring the inhibition of viral replication to levels undetectable by the most sensitive analytical methods; it prevents the development of drug resistance and enables the restoration of immune system function as well as preventing or delaying the occurrence of acquired immunodeficiency syndrome (AIDS). There are different therapeutic regimens based on the application of at least two nucleoside reverse transcriptase inhibitors (NRTIs) in combination with non-nucleoside reverse transcriptase inhibitors (NNRTIs), integrase transfer inhibitors (INSTIs), protease inhibitors (PIs), fusion inhibitors, and C-C Chemokine Receptor 5 (CCR5) antagonists. In addition to two NRTIs, recommended regimens include protease inhibitors (PIs) or integrase transfer inhibitors (INSTIs) [3,4,5].

Antiretroviral treatment effectively controls HIV infection and guarantees a good quality of life for many years. Despite its obvious benefits, HIV-infected people treated with antiretroviral therapy show a greater predisposition to metabolic disorders compared to the general population [4,5]. The prevalence of metabolic syndrome in people living with HIV (PLWH) is estimated at approximately 20–33% [6]. One of the causes of metabolic disorders in addition to lifestyle and the use of cART is the state of chronic inflammation. Increased expression of cytokines and other proinflammatory factors (e.g., soluble CD163, CD40, CD27, interleukin-6, CRP, D-dimer, cystatin C) especially in the area of adipose tissue, liver, skeletal muscles, and the digestive system causes changes in metabolism and increased storage of adipose tissue, predisposing to type two diabetes mellitus (T2DM) and cardiovascular diseases (CVD) [7]. However, the exact mechanisms behind the development of metabolic disorders in HIV-infected people are still not fully understood.

IRS is an adipomyokine, which is a fragment of fibronectin type III domain-containing protein 5 (FNDC5/FRCP2/PeP) in the cell membrane, secreted mainly by skeletal muscles and visceral and subcutaneous adipose tissue [8]. Irisin up-regulates the metabolism of adipose tissue and thermogenesis and reduces the formation of new adipocytes [8,9].

MSTN, otherwise known as growth differentiation factor-8 (GDF-8), is a negative regulator of muscle growth and a member of the transforming growth factor β (TGF-β) superfamily. It is expressed in skeletal muscles, the heart muscle, and adipose tissue [10]. In vitro studies also indicated its role in adipogenesis, neuronal control in insulin resistance, and communication between muscles and adipose tissue [11].

GLP-1 is an incretin peptide whose main action is to increase the release of insulin from pancreatic β cells and inhibit the release of glucagon, and it has a hypoglycemic effect [12]. It also inhibits gastric secretion and intestinal motility, regulating appetite in a manner similar to PYY. GLP-1 analogues are widely used in the treatment of T2DM and obesity [13].

PYY is a neuropeptide that regulates the gut-brain axis and acts as a satiety factor, inhibiting gut motility, appetite, and further food intake. It also regulates carbohydrate metabolism. It is secreted by endocrine L cells of the gastrointestinal tract after ingestion, together with GLP-1 incretin, and by pancreatic endocrine cells and gastric intestinal neurons [14].

DPP-4 is a serine exopeptidase that cleaves GLP-1 shortly after it is secreted. It is a widespread enzyme present in many tissues including the liver, gut, placenta, lung, and kidney [15]. DPP-4 inhibitors are also widely used in the treatment of T2DM and show a number of other beneficial pleiotropic effects apart from their hypoglycemic effect, e.g., anti-inflammatory [16].

FETU-A is a multifunctional glycoprotein mainly synthesized in the liver and released into the bloodstream. Significant amounts of FETU-A are also synthesized in adipose tissue [17].

PTX 3, like the C-reactive protein, is an acute phase protein and a member of the pentraxin family that is released within damaged tissue in dendritic cells (DCs), monocytes, macrophages, fibroblasts, chondrocytes, adipocytes, epithelial cells, vascular endothelial cells, smooth muscle cells, mesangial cells, granulosa cells by pro-inflammatory cytokines (such as interleukin-1β and tumor necrosis factor α), and microorganisms. Elevated levels of PTX3 have been found in the course of sepsis and bacteraemia, cardiovascular diseases, and coronary artery disease [18,19].

SDF-1, or chemokine CXCL12, is a multifunctional protein and an endogenous ligand for the C-X-C motif chemokine receptor four (CXCR4) receptor, which is also a coreceptor for some HIV-1 strains present in many tissues including fibroblasts, osteoblasts, and endothelial cells [20].

RANTES, or CCL5, is a chemokine from the CC subfamily that is a strong inflammatory mediator with chemotactic properties for immune cells in the place of injury or infection. RANTES which is released from activated platelets mediates the retention of monocytes in the inflamed site of the epithelium and promotes platelet aggregation as well as the formation of atherosclerotic lesions. It also regulates the activity of T lymphocytes in atherosclerotic lesions, contributing to the progression of atherosclerosis [20,21].

The aim of the study was to assess the effect of cART on the level of selected parameters characterizing carbohydrate and lipid metabolism, cardiovascular diseases, and inflammation (IRS, MSTN, PYY, GLP-1, DPP-4, FETU-A, PTX3, SDF-1, and RANTES) in the plasma of HIV-infected subjects before and one year after the implementation of cART. The analyses took into account the influence of parameters characterizing the state of the immune system such as CD4+ T cells and CD8+ T cells count, HIV RNA viral load, and the type of antiretroviral treatment regimen used: PIs or INSTIs. The obtained data may enable the clinical usefulness of selected parameters and provide new information on the development of disorders accompanying HIV infection.

## 2. Materials and Methods

### 2.1. Institutional Review Board Statement

The study was conducted in accordance with the Declaration of Helsinki, and the protocol was approved by Ethics Committee of Wroclaw Medical University (KB-597/2019). Written informed consent was obtained from all participants.

### 2.2. Patient Characteristics

The study group consisted of HIV-infected men (*N* = 53) at a mean age of 34 years treated in the Center for Preventive and Therapeutic Infectious Diseases and Addiction Therapy in Wroclaw and in the Department of Infectious Diseases, Liver Diseases and Acquired Immune Deficiencies of the Medical University of Wroclaw. All patients were infected with the HIV-1 strain. Inclusion criteria for the study group were confirmation of the presence of HIV infection and intake of cART drugs. Exclusion criteria were diseases such as: diabetes mellitus, cancer, hypertension, urinary tract diseases, and concomitant use of drugs other than cART.

The control group consisted of 35 healthy HIV-negative men at a mean age of 36 years without any chronic or inflammatory diseases such as: diabetes mellitus, renal diseases, cardiovascular diseases, or hepatitis B or C virus infection.

HIV-infected men were treated with two therapeutic regimens which included two NRTIs (emtricitabine and tenofovir alafenamide) in combination with PIs (ritonavir-boosted lopinavir or cobicistat-boosted darunavir) or INSTIs (dolutegravir).

Data on CD4+ T cells and CD8+ T cells count, HIV RNA viral load, and biochemical parameters such as TC (total cholesterol), LDL-C (LDL cholesterol), HDL-C (HDL cholesterol), TG (triglycerides), FBG (fasting blood glucose), and BMI (body mass index) were obtained from medical records of patients.

### 2.3. Determination of Selected Parameters Levels in Plasma of HIV-Infected Men and Healthy Controls

In HIV-infected men, blood was drawn twice: before and one year after starting to take cART. Whole human blood was collected from HIV-infected men and the control in a fasting state. Blood samples were taken into EDTA-treated tubes (5 mL blood, containing 1.6 mg/mL EDTA, Sarstedt, Poland). Tubes were centrifuged by MPW-350 laboratory centrifuge (MPW Instruments, Warszawa, Poland) at 1500× *g* for 10 min to separate the plasma. Plasma was removed and placed in Eppendorf tubes (Eppendorf AG, Hamburg, Germany) and stored at −80 °C for further investigation.

The measurement of concentrations of selected parameters was performed by enzyme-linked immunoassay (ELISA) method using: human irisin ELISA kit (Cat.No E3253Hu), human growth differentiation factor 8 ELISA kit (Cat.No E3058Hu), human peptide YY ELISA kit (Cat.No E1369Hu), human glucagon-like peptide 1 ELISA kit (Cat.No E0022Hu), human dipeptidyl peptidase 4 ELISA kit (Cat.No E6631Hu), human fetuin A ELISA kit (Cat.No E1386Hu), human pentraxin 3 ELISA kit (Cat.No E1938Hu), human stromal cell derived factor 1 ELISA kit (Cat.No E3353Hu), and human regulated on activation in normal T cell expressed and secreted or C-C motif chemokine 5 ELISA kit (Cat.No E3663Hu), Bioassay Technology Laboratory (BT Lab; Shanghai Korain Biotech Co., Ltd., Shanghai, China) according to the manufacturer’s instructions. Standards and serum samples were added into a 96-well plate. After adding biotin-conjugated antibodies and streptavidin–horseradish peroxidase, the plate was incubated for 60 min at 37 °C. The wells were then washed five times with wash buffer. Substrate solutions A and B were added and the plate was incubated for 10 min at 37 °C for color development. Finally, the reaction was stopped by the stop solution. The intensity of the color in each well was measured in a microplate reader. Absorbance was read at 450 nm with a microplate reader STAT FAX 2100 (Awareness Technology Inc., Palm City, FL, USA).

### 2.4. Statistical Analysis

Statistical analysis was performed using Statistica 13.3 PL (StatSoft, Cracow, Poland). All investigated quantitative variables were checked with the Shapiro–Wilk test to establish the type of distribution. Variables with non-parametric distribution were presented as the median and interquartile range (IQR) 25–75%. The comparison of quantitative variables between the groups was performed using the Kruskal–Wallis test and the U Mann–Whitney test. The Kruskal–Wallis test was used to compare three groups (HIV-infected men before cART, after cART, and controls) in terms of the quantitative variables studied. The statistically significant result of the Kruskal–Wallis test indicated that at least one group differs from the other group. Therefore, a post-hoc test (Dunn test with Bonferroni correction) was then performed to see exactly which groups differed from each other. Within-group comparison between results before and after cART was made using the Wilcoxon test. HIV-infected men were also analyzed in subgroups depending on the number of CD4+ T cells count (below and above 300 cells/µL), CD8+ T cells count (below and above 1000 cells/µL), HIV RNA viral load (below and above 100,000 RNA copies/mL), and the type of therapeutic regimen (INSTIs or PIs). For all analyses, *p* < 0.05 was accepted as a significant value.

## 3. Results

The study groups consisted of HIV-infected men before cART (A) and one year after the implementation of cART (B), and a control group (C). Demographic and biochemical data (BMI, FBG, TG, LDL-C, HDL-C, and TG) are presented in Table 1.

The median values of age, BMI, FBG, TG, LDL-C, HDL-C, and TG in HIV-infected men before (A) and after treatment (B) were similar in the study groups of patients and the control group and were not statistically significant (*p* > 0.05).

Immunological data concerning the groups of patients before (A) and after cART (B) are provided in Table 2.

The difference in the median values of HIV RNA viral load and CD4+ T cells and CD8+ T cells count in HIV-infected men before (A) and after treatment (B) were statistically significant.

### 3.1. Panel of Selected Parameters in Plasma of HIV-Infected Men before cART Therapy, after cART Therapy, and in the Control Group with Statistical Analysis

Values of IRS, MSTN, PYY, DPP-4, FETU-A, PTX3, SDF-1, and RANTES before (A) and after (B) cART in the group of HIV-infected men and the control group (C) with statistical analysis are provided in Table 3.

The analysis of examined parameters in HIV-infected men before and after cART and the control group showed statistically significant differences for the following parameters: IRS, MSTN, PYY, GLP-1, PTX 3, and RANTES. No significant differences were found for three parameters: DPP-4, FETU-A, and SDF-1.

For IRS, MSTN, PYY, GLP-1, PTX 3, and RANTES in HIV-infected men before cART (A), the median levels were significantly lower compared to the median levels obtained in the control group (2.4-, 2-, 1.2-, 2.9-, 1.6-, and 2.3-fold, respectively). The median levels of IRS, MSTN, GLP-1, and FETU-A in HIV-infected men before cART (A) were lower than the medians obtained in HIV-infected men after cART (B), but the differences were not statistically significant. The median levels of DPP-4, PTX 3, and RANTES in HIV-infected men before cART (A) were higher than the medians obtained in HIV-infected men after cART (B), but the differences were not statistically significant. The median level for SDF- 1 was the same in HIV-infected men both before and after cART. The only parameter whose median level was statistically significantly higher (two-fold) in HIV-infected men before (A) compared to after cART (B) was PYY (Appendix A).

### 3.2. Results of Selected Parameters before and after cART in HIV-Infected Men by CD4+ T and CD8+ T Cells Counts

The values of selected parameters in HIV-infected men both before cART (A) and after treatment (B) depending on the CD4+ T cells count (below and above 300 cells/µL) and the CD8+ T cells count (below and above 1000 cells/µL) at the time of sampling, are presented in Table 4.

Median levels of the studied parameters slightly differed in HIV-infected men with CD4+ T cells ≤300 and CD4+ T cells >300 before (A) and after (B) cART.

In HIV-infected men with CD8+ T cells ≤1000 before cART (A), an increase (almost two-fold) in the median level was observed for all parameters except PYY (3.5-fold decrease) compared to the median level of these parameters after cART. Significant statistical differences were shown only for FETU-A (*p* = 0.03) and PTX-3 (*p* = 0.04).

In HIV-infected men with CD8+ T cells >1000 count, an increase in the median level was observed for all parameters except PYY, DPP-4, and PTX-3 (decrease) compared to the median levels of these parameters after cART.

### 3.3. Results of Selected Parameters before and after cART in HIV-Infected Men by HIV RNA Viral Load

The concentration and the results of statistical analysis of selected parameters in HIV-infected men before cART (A) depending on the amount of HIV RNA copies (below and above 100,000 copies/mL) at the time of sampling are presented in Table 5.

None of the HIV-infected men with a pre-treatment viral load HIV RNA > 100,000 (copies/mL) maintained such a load after cART (B).

Median levels of all parameters in HIV-infected men with HIV RNA viral load HIV RNA ≤ 100,000 (copies/mL) were higher compared to the results obtained in HIV-infected men with HIV RNA viral load HIV RNA > 100,000 (copies/mL).

### 3.4. Examined Parameters in HIV-Infected Men Subjected to cART with Protease Inhibitors (PIs) and Integrase Transfer Inhibitors (INSTIs) Treatment

Medians and interquartile ranges for selected parameters in HIV-infected men after cART subgroups with protease inhibitors (PIs) and integrase transfer inhibitors (INSTIs) treatment are presented in Table 6.

The antiretroviral regimen used, whether cART with protease inhibitors or integrase transfer inhibitors treatment, had no significant effect on the selected parameters’ levels.

The median levels of the parameters: IRS, MSTN, PPY, GLP-1, DPP-4, FETU-A, PTX 3, SDF-1, and RANTES were lower in HIV-infected men treated with PIs compared to HIV-infected men treated with INSTIs (3.6-, 3-, 3.7-, 4.2-, 5.6-, 5.7-, 4.2-, 4.7-, 3.7-fold, respectively).

## 4. Discussion

In the era of effective cART therapy which has significantly extended the life expectancy of HIV-infected patients, the priority is to improve the quality of life of infected people by optimizing cART therapy. Literature data indicate an increased risk of metabolic disorders in HIV-infected subjects who use cART [22,23]. However, the exact mechanisms of these changes are still not sufficiently understood and there are no precisely defined parameters that can describe these disorders. In the presented study, the level of nine parameters related to carbohydrate and lipid metabolism and inflammation were measured. The abovementioned parameters are some of the main factors contributing to the development of concomitant diseases, especially metabolic diseases, in HIV-infected patients. The levels of selected parameters (IRS, MSTN, PYY, GLP-1, DPP-4, FETU-A, PTX 3, SDF-1, and RANTES) were compared, both before and one year after the cART application, depending on the value of individual parameters characterizing the clinical status of patients (CD4+ T cells and CD8+ T cell count and HIV RNA viral load) and the type of treatment used (PIs or INSTIs). There are no studies similar to our investigation in the scientific literature, which has made the subject of particular interest.

The analysis of HIV-infected men before and after cART and the control group showed statistically significant differences for the following parameters: IRIS, MSTN, PYY, GLP-1, PTX 3, and RANTES, but no significant differences were found for DPP-4, FETU-A, and SDF-1.

Data in the literature indicate a protective function of IRS in the development of diseases such as: obesity, insulin resistance, type two diabetes, cardiovascular diseases, or cancer due to its anti-inflammatory activity and the effect of increasing insulin sensitivity, glycogenesis, and inhibiting gluconeogenesis [24]. IRS levels have been shown to be elevated in obese pre-diabetic subjects and reduced by about 40% in people with T2DM [25]. Sesti et al. [26] showed that in healthy non-diabetic subjects, IRS levels were positively correlated with body fat mass and insulin levels and negatively correlated with insulin-stimulated glucose disposal and insulin clearance, which as the authors concluded, may be associated with compensatory actions in the development of metabolic disorders [26]. Similar correlations between IRS concentration and the parameters of carbohydrate metabolism were noticed by Trombeta et al. [27] and Moreno-Perez et al. [28] in HIV-infected men. However, this study examined plasma IRS levels in both cART-treated and untreated patients, making it impossible to assess the effects of cART [27,28]. Srinivasa et al. [29] measured IRS levels in HIV-infected people with established metabolic syndrome as defined by the National Cholesterol Education Program (NCEP). The authors showed that the level of IRS was statistically significantly lower in the group of HIV-infected patients compared to the control group (*p* = 0.003). In our own study, plasma median levels of IRS were also statistically significantly lower in HIV-infected men before cART compared to the control group (*p* = 0.02). Despite the lack of statistical significance between the groups before and after cART, an upward trend in the IRS median level after cART could be observed, which may indicate a negative impact of HIV infection on IRS levels as well as a beneficial effect of cART. However, compared to our studies, in the studies conducted by Srinivasa et al. [29], patients used cART for over six years on average, so it is possible that the follow-up was too short for us to notice the above dependence or correlations in our own study. It is possible that in the following years of cART use, a further reduction in the level of circulating IRS is observed as a result of the development of metabolic disorders. We have found no other studies showing IRS levels in HIV-infected patients not treated with cART, which would have allowed the effect of HIV infection alone on IRS levels to be assessed.

In addition to preventing muscle hypertrophy, MSTN has also been shown to induce insulin resistance in a mechanism dependent on nuclear factor kappa-light-chain-enhancer of activated B cells (NF-κB) and SMAD family member 3 (SMAD3) [10,11]. The concentration of MSTN in the serum depends on nutritional status—reduced levels have been found in people suffering from anorexia nervosa while increased levels have been observed in obese people or people with metabolic syndrome, T2DM, or pre-diabetes [30,31]. Assyov et al. [11] revealed that MSTN serum levels in people with normoglycemia had the lowest levels of MSTN compared to people with T2DM and pre-diabetes. A positive correlation with FBG (fasting blood glucose) and the homeostatic model of assessment for insulin resistance (HOMA-IR) was demonstrated [11]. Lower serum MSTN levels were also observed in people with metabolic syndrome, central obesity, and higher TG and lower HDL-C levels [32]. In our study, no changes in FBG were found before or after cART in HIV-infected men. These levels were also similar to the results in the control group. The MSTN median level was the lowest in HIV-infected men before cART compared to the control group, and this difference was statistically significant (*p* = 0.02). Other results were obtained by Gonzalez-Cadavid et al. [33], who showed that serum levels of MSTN in HIV-infected men were statistically significantly higher compared to the control group. However, the studied patients were treated with cART and had significant weight loss and AIDS wasting syndrome [33]. The above data may indicate changes in the MSTN level during HIV infection that is dependent on the clinical condition of the patient, and the analysis of the MSTN level alone may not be sufficient. There is also still a need for research on the clinical utility of measuring MSTN in the assessment of comorbidities in HIV-infected individuals.

It has been shown that postprandial secretion of intestinal incretin hormones, mainly GLP-1, is reduced in people with T2DM and weakened in obese people with insulin resistance and normal glucose levels [34]. Andersen et al. [35] controversially showed increased GLP-1 secretion in response to insulinotropic stimuli in HIV-infected patients with impaired glucose tolerance compared to HIV-infected patients with normal glucose tolerance. The authors pointed to the potential existence of a compensatory mechanism as a result of developing insulin resistance [35]. As a result of chronic immune activation and damage to enteroendocrine cells in the course of HIV infection and the development of HIV-associated enteropathy, incretin release is impaired in infected individuals [36]. This was confirmed by our own research in which median GLP-1 was significantly lower in HIV-infected men compared to the control group (*p* = 0.004). Our own research also showed changes in the median level of GLP-1 in the course of HIV infection and an upward trend one year after cART, which confirms earlier data on the influence of infection on the secretion of incretins.

PYY is metabolized through DPP-4 to its active form. Due to the action of PYY on the pancreatic β-cell NPYR1 receptor, it is considered a therapeutic target for antidiabetic therapies [37]. A positive correlation of the PYY level with hs-CRP has also been demonstrated, indicating the inflammatory regulation of the secretion of this neuropeptide. In addition, a correlation with the PYY level has been demonstrated for cardiovascular risk factors (diabetes, hypertension, and hypercholesterolemia) and cardiovascular events [38]. Our own study showed a higher median level of PYY in HIV-infected men before cART compared to after cART (*p* = 0.04) and a lower level of PYY than in the control group (*p* = 0.02). Differences in the median level of PYY before and after cART may be the result of chronic inflammation during infection or the abovementioned HIV-associated enteropathy. These data indicate a multidirectional effect of PYY, and our own studies confirmed the need for further research on the role of changes in GLP-1 or PYY secretion in the course and pathogenesis-associated disorders of HIV.

DPP-4 has been shown to have multidirectional immunomodulatory activity; it regulates the functions of, among others, CD4+ T lymphocytes, natural killer cells, and macrophages [16]. Levels of GLP-1 and PYY are closely related to the activity of DPP-4 peptidase which causes their enzymatic degradation. Hosono et al. [39] showed no statistically significant differences in the plasma levels of DPP-4 in HIV-infected people compared to uninfected controls. However, DPP-4 activity was significantly lower in infected individuals (*p* < 0.0001) and correlated positively with CD4+ T and CD8+ T cell count while correlating inversely with HIV RNA [39]. Songok et al. [40] showed that DPP-4 levels were not significantly different in the HIV-positive group compared to the HIV-negative controls [40]. Similarly in our own study, no statistically significant differences in median DPP-4 levels were found. However, the authors demonstrated that significantly higher plasma levels of DPP-4 occurred in people exposed but resistant to HIV infection. The authors showed that in people resistant to infection, the expression of DPP-4 in CD4+ T cells was significantly increased compared to HIV-negative and unexposed people (*p* = 0.0003) [40]. Opposite results were shown in our own studies, where the median plasma level of DPP-4 was the highest in HIV-infected patients and decreased after the administration of the therapy. These data may indicate a lack of correlation between enzymatic activity, expression in CD4+ T cells, and the plasma level of DPP-4 in the fasting state.

It has been shown that high levels of FETU-A can be considered a risk factor for insulin resistance, T2DM, CVD by inhibition of adiponectin expression, peroxisome proliferator-activated receptor-γ (PPARγ) or activating the toll-like receptor 4 (TLR-4), and as a factor reducing the risk of coronary artery disease [41]. The level of FETU-A, as a negative acute phase protein, decreases in the course of inflammation and increases again after recovery [41]. FETU-A may therefore be useful in monitoring inflammation in HIV patients. In our study, median plasma levels of FETU-A did not differ significantly in HIV-infected men before and after cART and in the control group. However, the median level of FETU-A was the lowest in the group before cART and the highest in the control group, which confirms that the changes in the FETU-A level were a result of the development of inflammation that is characteristic of HIV infection [7]. Statistically significant lower plasma FETU-A levels in HIV-infected men with CD8+ T-cell count ≤1000 cells/µL before cART compared to after cART with CD8+ T-cell count ≤1000 cells/µL (*p* = 0.03) were also observed. The obtained results indicate that the level of FETU-A in HIV-infected men increased as a result of the applied treatment. However, more research is needed to determine the usefulness of FETU-A in monitoring the course of HIV infection and the effectiveness of therapy.

PTX 3 is considered to be an independent marker in the diagnosis of coronary artery disease, and its concentration is negatively correlated with the course of the disease. The concentration of PTX 3 increases in asymptomatic atherosclerosis or coronary diseases and acute coronary syndrome [18]. It has also been shown to have a prognostic value in the course of some viral and bacterial infections [42,43]. To the best of our knowledge, there are no other studies examining the PTX 3 level in HIV-positive patients. Our research yielded unexpected results. The PTX 3 median level was statistically significantly higher in the control group compared to HIV-infected men before cART (*p* = 0.03). These data may suggest the existence of separate mechanisms of PTX 3 release impairment due to HIV infection, indicating the need for further research in this aspect. Interesting results were also obtained by analyzing the median PTX 3 level in subgroups depending on CD8+ T-cell count. Statistically significant lower median plasma levels of PTX 3 were found in the group of HIV-infected men before cART with CD8+ T cell count ≤1000 cells/µL compared to after cART with CD8+ T cell count ≤1000 cells/µL. As a result of chronic inflammation in the course of HIV, there was excessive activation of CD8+ T cells as a result of the action of pro-inflammatory cytokines. These data may confirm previous reports, which showed that there is insufficient effectiveness of cART in normalizing CD8+ T cell count and intensified inflammatory processes, even in patients treated with cART [44].

In our own study, the median SDF-1 level did not differ significantly between HIV-infected men before cART, after cART, and in the control group. Similarly Yeregui et al. [23] did not find statistically significant differences in SDF-1 level of HIV-infected individuals starting cART with CD4+ T cell count ≤200 cells/µL compared to patients starting cART with CD4+ T cell count >200 cells/µL. However, the authors achieved statistically significant higher levels of SDF-1 in patients starting cART with CD4+ T cell count ≤200 cells/µL, whom after one year of cART achieved no more than CD4+ T cell count <250 cells/µL (immunological nonrecovery) compared to those who attained immunological recovery, indicating a prognostic value in poor immunological recovery in response to cART. However, the authors did not compare SDF-1 levels to the control group or SDF-1 levels before and after cART. In our own study, no significant differences were found between the level of SDF-1 in subgroups depending on CD4+ T cell count (≤300 cells/µL and >300 cells/µL) in HIV-infected men before and after cART. However, further studies are needed to confirm the prognostic role in poor immunological recovery for SDF-1 in patients with lower CD4+ T cell count.

Elevated plasma levels of RANTES have been shown to correlate with the progression of coronary artery disease and acute coronary syndrome. In healthy people, they were prognostic indicators of the metabolic syndrome [23]. Increased levels of RANTES have also been demonstrated in diseases with chronic immune activation [24]. RANTES controls HIV infection through its chemotactic properties for immune cells and by limiting the interaction of the virus with the CCR5 coreceptor [45,46]. In our study, the median plasma level of RANTES in HIV-infected men before cART was statistically significantly lower compared to the control group (*p* = 0.02), which may indicate a weakened immune response in the course of infection or impaired CD8+ cell function. Further research is necessary to provide information on changes in RANTES levels in subsequent years of cART use and to determine the usefulness of this parameter in monitoring the course of HIV infection or treatment effectiveness.

### Limitations

Our study had some limitations. Due to the too short (one year) follow-up, no significant changes in the basic biochemical parameters such as FBG, TG, LDL-C, HDL-C, and BMI were observed. It is possible that the exposure time to the drug was too short to cause the changes due to the use of cART that are widely described in the literature [3,4,5]. Therefore, in the above study, changes in the median levels of parameters before cART were mainly observed compared to the control group. It was also not possible to test some parameters after nutritional stimuli (GLP-1 or PYY) or after physical activity (IRS), or measure the activity enzyme (DPP-4). Too short of a follow-up period likely prevented significant changes in the levels of the studied parameters as a result of cART from showing, depending on the therapeutic regimen used (PIs or INSTIs). However, it was noted that the median levels of all parameters tested in patients treated with INSTIs were higher than in patients treated with PIs, providing an interesting area for future research to be continued in a long-term follow-up.

## 5. Conclusions

Statistically significant reduced median levels of IRS, MSTN, PYY, GLP-1, PTX 3, and RANTES were demonstrated in HIV-infected men before cART compared to the control group. Statistically significant differences were also demonstrated in the median levels of FETU-A and PTX 3 between HIV-infected men before cART with CD8+ T cells count ≤1000 cells/µL and after cART with CD8+ T cells count ≤1000 cells/µL. Higher median levels of the tested parameters were also noted in men treated with INSTIs compared to PIs. The obtained results indicated significant changes in the levels of selected parameters as a result of HIV infection and changes along with the implementation of cART even in a relatively short observation period of one year, which indicated the need for further research on the diagnostic and prognostic value of the parameters. The obtained data may be useful in monitoring the course of HIV infection, treatment efficacy, or optimization of therapy. This may be helpful in the prevention or development of comorbidities associated with HIV infection and chronic cART use.

The obtained results may also indicate the need for further research in terms of the role of the studied parameters in the pathogenesis of disorders comorbid with HIV infection.

## Figures and Tables

**Table 1 jcm-11-01713-t001:** Demographic and biochemical data of HIV-infected men before (A) and after cART (B) and the control group (C) with statistical analysis.

Group Characteristics	A	B	C	*p* *
	Me (IQR)	Me (IQR)	Me (IQR)	
Age (Y)	33 (28–40)	34 (29–41)	36 (30–43)	0.76
BMI [kg/m^2^]	24.15 (21.55–24.80)	24.00 (21.56–24.81)	22.30 (18.00–26.80)	0.68
FBG [mg/dL]	96.00 (91.90–98.10)	98.90 (92.40–103.90)	95.00 (87.20–99.20)	0.08
TC [mg/dL]	174.90 (156.00–187.00)	174.00 (155.00–189.00)	180.00 (165.00–195.00)	0.44
LDL-C [mg/dL]	96.00 (85.00–103.00)	99.06 (80.00–110.00)	100.00 (96.00–115.00)	0.11
HDL-C [mg/dL]	76.60 (63.00–84.00)	77.15 (67.00–82.00)	75.00 (59.00–101.00)	0.15
TG [mg/dL]	149.00 (119.00–160.00)	157.20 (100.00–162.00)	163.00 (151.00–175.00)	0.35

Abbreviation: A—HIV-infected men before cART; B—HIV-infected men after cART; C—control group; BMI—body mass index; FBG—fasting blood glucose; TC—total cholesterol; LDL-C—LDL cholesterol; HDL-C—HDL cholesterol; TG—triglycerides; Me—median; IQR—interquartile range; and N—number of participants. * *p*—statistical significance by Kruskal–Wallis test.

**Table 2 jcm-11-01713-t002:** Immunological data of HIV-infected men groups before (A) and after cART (B) with statistical analysis.

Group Characteristics	A	B	*p* *
	Me (IQR)	Me (IQR)	
CD4+ T [cells/µL]	340 (234–386)	570 (398–762)	<0.001
CD8+ T [cells/µL]	999 (717–1190)	855 (706–1062)	0.004
HIV RNA [copies/mL]	148,000 (5190–245,000)	20 (15–34)	<0.001

Abbreviation: A—HIV-infected men before cART; B—HIV-infected men after cART; C—control group; Me—median; IQR—interquartile range; and N—number of participants. * *p*—statistical significance by Wilcoxon test.

**Table 3 jcm-11-01713-t003:** Results for selected parameters in the plasma of HIV-infected men before (A) and after cART (B) and in the control group (C) with statistical analysis.

Groups	A	B	C	*p* *	Post-Hoc
	Me (IQR)	Me (IQR)	Me (IQR)		
**IRS** **[ng/mL]**	4.70 (1.60–21.80)	8.90 (2.50–41.30)	11.30 (3.20–38.30)	0.02	A:C = 0.02 B:C = 0.38 A:B = 0.13
**MSTN** **[ng/mL]**	162.80 (80.50–542.00)	253.00 (105.20–2060.50)	318.40 (129.40–1575.00)	0.03	A:C = 0.02 B:C = 0.23 A:B = 0.25
**PYY** **[pg/mL]**	156.30 (126.90–286.40)	79.20 (31.40–262.60)	181.50 (43.80–446.40)	0.03	A:C = 0.02 B:C = 0.30 A:B = 0.04
**GLP-1** **[ng/mL]**	209.60 (84.80–604.60)	263.40 (132.00–2516.00)	607.10 (184.60–1440.00)	0.01	A:C = 0.004 B:C = 0.24 A:B = 0.19
**DPP-4** **[ng/mL]**	239.80 (181.80–430.40)	109.00 (51.30–1296.50)	174.30 (66.70–1260.00)	0.17	A:C = 1.00 B:C = 0.45 A:B = 0.33
**FETU-A** **[ng/mL]**	265.2 (132.30–1387.00)	363.90 (142.00–3568.50)	620.60 (157.50–2302.00)	0.12	A:C = 0.10 B:C = 0.58 A:B = 0.58
**PTX 3** **[ng/mL]**	2.70 (1.10–7.90)	2.60 (1.30–36.10)	4.40 (1.50–17.40)	0.03	A:C = 0.04 B:C = 0.53 A:B = 0.13
**SDF-1** **[ng/mL]**	1.7 (0.70–6.40)	1.70 (0.80–18.80)	3.30 (1.10–8.50)	0.19	A:C = 0.10 B:C = 0.41 A:B = 1.0
**RANTES** **[ng/mL]**	330.50 (107.90–782.40)	317.90 (165.90–3129.00)	750.90 (182.60–2625.00)	0.02	A:C = 0.02 B:C = 0.42 A:B = 0.15

Abbreviations: IRS—irisin; MSTN—myostatin; PYY—peptide YY; GLP-1—glucagon-like peptide-1; DPP-4—dipeptidyl peptidase IV; FETU-A—fetuin A; PTX3—pentraxin 3; SDF-1-1—chemokine stromal cell-derived factor 1 regulated on activation; RANTES—normal T cell expressed and secreted; A—HIV-infected men before cART; B—HIV-infected men after cART; C—control group; Me—median; and IQR—interquartile range. * *p*—statistical significance by Kruskal–Wallis test.

**Table 4 jcm-11-01713-t004:** Results of selected parameters before and after cART in HIV-infected men by CD4+ T cells count (below and above 300 cells/µL) and by CD8+ T cells count (below and above 1000 cells/µL) with statistical analysis.

	CD4+ T Cells ≤ 300 [Cells/µL]		*p* *	CD8+ T Cells ≤ 1000 [Cells/µL]		*p* *
	Me (IQR)			Me (IQR)		
	A (*N* = 18)	B (*N* = 10)		A (*N* = 26)	B (*N* = 36)	
**IRS ** **[ng/mL]**	4.55 (1.70–22.40)	4.70 (1.60–18.10)	0.44	5.20 (1.70–32.80)	10.00 (2.50–75.70)	0.09
**MSTN ** **[ng/mL]**	149.10 (80.50–903.90)	162.80 (82.40–447.60)	0.07	273.00 (80.50–572.80)	484.70 (127.00- 2960.00)	0.06
**PYY ** **[pg/mL]**	161.75 (131.40–334.50)	156.30 (125.90–255.60)	0.21	173.20 (126.90–293.20)	158.60 (39.30–611.40)	0.45
**GLP-1 ** **[ng/mL]**	206.90 (100.10–697.60)	209.60 (82.10–456.20)	0.60	265.65 (83.00–499.30)	559.60 (141.10–3598.00)	0.10
**DPP-4 ** **[ng/mL]**	245.10 (190.20–488.40)	239.80 (181.95–341.65)	0.68	245.10 (174.50–320.00)	305.20 (54.00–1452.00)	0.45
**FETU-A ** **[ng/mL]**	269.10 (136.20–1487.00)	265.15 (119.05–820.75)	0.68	344.20 (132.30–1068.00)	1112.10 (173.50–4109.00)	0.03
**PTX3 ** **[ng/mL]**	2.80 (1.10–8.50)	2.70 (1.00–5.40)	0.69	2.85 (1.10–5.30)	7.50 (1.50–43.70)	0.04
**SDF-1 ** **[ng/mL]**	1.90 (0.70–9.60)	1.65 (0.70–4.90)	0.59	2.35 (0.70–5.010)	4.20 (1.10–41.10)	0.06
**RANTES ** **[ng/mL]**	346.40 (127.10–1075.00)	330.50 (106.30–767.20)	0.68	434.70 (127.10–767.20)	816.30 (196.10–4804.00)	0.11
	**CD4+ T cells > 300** **[cells/µL]**		***p* ***	**CD8+ T cells > 1000** **[cells/µL]**		***p* ***
	**Me ** **(IQR)**			**Me** **(IQR)**		
	**A ** **(*N* = 35)**	**B ** **(*N* = 43)**		**A ** **(*N* = 27)**	**B ** **(*N* = 17)**	
**IRS ** **[ng/mL]**	9.50 (2.20–15.30)	9.40 (2.50–49.10)	0.10	2.30 (1.60–18.40)	5.850 (2.25–33.40)	0.15
**MSTN ** **[ng/mL]**	322.55 (103.40–770.50)	299.50 (105.20–2684.00)	0.30	122.10 (78.50–903.90)	221.10 (107.90–1856.00)	0.20
**PYY ** **[pg/mL]**	101.70 (32.80–217.20)	83.80 (30.10–452.90)	0.27	155.00 (125.90–286.40)	43.70 (24.70–280.50)	0.86
**GLP-1 ** **[ng/mL]**	408.40 (124.30–661.80)	340.20 (133.90–3170.00)	0.23	103.20 (84.80–697.60)	171.70 (133.90–2360.00)	0.06
**DPP-4 ** **[ng/mL]**	176.75 (48.30–512.10)	123.50 (52.40–1356.00)	0.95	229.30 (181.80–430.40)	58.50 (51.90–1356.00)	0.57
**FETU-A ** **[ng/mL]**	722.95 (160.50–1433.00)	454.30 (133.30–4109.00)	0.24	164.90 (122.50–1586.00)	209.00 (158.00–3290.00)	0.08
**PTX3 ** **[ng/mL]**	3.25 (1.20–8.90)	2.90 (1.30–39.50)	0.15	2.40 (1.00–9.60)	1.60 (1.20–32.20)	0.11
**SDF-1 ** **[ng/mL]**	2.40 (0.70–5.20)	1.90 (0.80–23.70)	0.14	0.90 (0.70–7.70)	0.90 (0.80–17.80)	0.12
**RANTES ** **[ng/mL]**	540.45 (166.60–999.10)	361.40 (165.10–3581.00)	0.12	176.40 (92.30–1075.00)	192.30 (169.30–3436.00)	0.06

Abbreviation: A—HIV-infected man before cART; B—HIV-infected man after cART; Me—median; IQR—interquartile range; *N*—number of participants; IRS—irisin; MSTN—myostatin; PYY—peptide YY; GLP-1—glucagon-like peptide-1; DPP-4—dipeptidyl peptidase IV; FETU-A—fetuin A; PTX3—pentraxin 3; SDF-1—chemokine stromal cell-derived factor 1; and RANTES—regulated on activation, normal T cell expressed and secreted. * *p*—statistical significance by Wilcoxon test.

**Table 5 jcm-11-01713-t005:** Results of selected parameters before and after cART in HIV-infected men by HIV RNA viral load (below and above 100,000 copies/mL) with statistical analysis.

	HIV RNA ≤ 100,000 [Copies/mL] (*N* = 31)	HIV RNA > 100,000 [Copies/mL] (*N* = 22)	*p* *
	Me (IQR)	Me (IQR)	
	HIV-Infected Men before cART (A)		
**IRS** **[ng/mL]**	5.15 (1.60–21.80)	4.10 (1.80–18.10)	0.24
**MSTN** **[ng/mL]**	261.60 (83.20–707.60)	128.75 (72.90–445.50)	0.36
**PYY** **[pg/mL]**	192.05 (125.85–313.85)	149.35 (126.90–255.60)	0.87
**GLP-1** **[ng/mL]**	243.95 (94.15–684.85)	103.20 (79.40–369.90)	0.89
**DPP-4** **[ng/mL]**	276.45 (183.00–460.05)	229.30 (179.70–281.30)	0.47
**FETU-A** **[ng/mL]**	287.95 (133.30–1437.00)	156.00 (121.50–458.30)	0.56
**PTX3** **[ng/mL]**	3.00 (1.10–8.45)	2.40 (0.90–4.20)	0.57
**SDF-1** **[ng/mL]**	2.00 (0.70–8.65)	0.90 (0.70–2.80)	0.50
**RANTES** **[ng/mL]**	386.45 (117.50–992.45)	207.80 (88.45–608.55)	0.72

Abbreviation: A—HIV-infected man before cART; B—HIV-infected man after cART; Me—median; IQR—interquartile range; *N*—number of participants; IRS—irisin; MSTN—myostatin; PYY—peptide YY; GLP-1—glucagon-like peptide-1; DPP-4—dipeptidyl peptidase IV; FETU-A—fetuin A; PTX3—pentraxin 3, SDF-1—chemokine stromal cell-derived factor 1; and RANTES—regulated on activation, normal T cell expressed and secreted. * *p*—statistical significance by U Mann–Whitney test.

**Table 6 jcm-11-01713-t006:** Results of selected parameters in HIV-infected men after cART in the subgroup with protease inhibitors (PIs) treatment and the subgroup with integrase transfer inhibitors (INSTIs) treatment with statistical analysis.

Parameters	PIs (*N* = 25)	INSTIs (*N* = 28)	*p* *
	Me (IQR)	Me (IQR)	
	HIV-Infected Men after cART (B)		
**IRS** **[ng/mL]**	2.90 (2.30–15.90)	10.60 (2.90–77.40)	0.18
**MSTN** **[ng/mL]**	160.00 (105.20–917.90)	484.70 (110.60–3589.00)	0.20
**PYY** **[pg/mL]**	54.50 (32.60–205.50)	201.20 (30.10–666.60)	0.08
**GLP-1** **[ng/mL]**	159.30 (124.30–709.20)	661.80 (150.00–3484.00)	0.09
**DPP-4** **[ng/mL]**	70.90 (48.30–541.80)	395.80 (52.60–1452.00)	0.14
**FETU-A** **[ng/mL]**	224.25 (142.80–1433.00)	1267.00 (173.50–5173.00)	0.33
**PTX3** **[ng/mL]**	1.95 (1.20–9.80)	8.10 (1.50–43.70)	0.31
**SDF-1** **[ng/mL]**	1.15 (0.70–5.20)	5.20 (0.80–41.10)	0.22
**RANTES** **[ng/mL]**	240.40 (155.40–999.10)	893.90 (169.30–4804.00)	0.24

Abbreviation: cART—combined antiretroviral therapy; B—HIV-infected man after cART; PIs—protease inhibitors treatment; INSTIs—integrase transfer inhibitors; Me—median; IQR—interquartile range; *N*—number of participants; NS—not statistically significant; IRS—irisin; MSTN—myostatin; PYY—peptide YY; GLP-1—glucagon-like peptide-1; DPP-4—dipeptidyl peptidase IV; FETU-A—fetuin A; PTX 3—pentraxin 3; SDF-1—chemokine stromal cell-derived factor 1; and RANTES—regulated activation normal T cell expressed and secreted. * *p*—statistical significance by Wilcoxon test.

## Data Availability

Not applicable.

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
