# Peer review of "Effect of Combined Antiretroviral Therapy on the Levels of Selected Parameters Reflecting Metabolic and Inflammatory Disturbances in HIV-Infected Patients"

_jcm, 2022, doi:10.3390/jcm11061713_

Round 1

Reviewer 1 Report

This manuscript describes the changes of selected parameters related to inflammation and metabolic disturbances during HIV infection and therapy. Although this is a purely descriptive study, the results presented in this manuscript may be of interest for the scientists involved in this field and may lead to further mechanistic studies.

This manuscript suffers of some weaknesses and some points must be clarified.

Minor points:

  1. The abstract mentions statistically significant differences without any detail (line 22). Some numbers should be given for those significant differences: median (IQ) and p value.
  2. The introduction appears a little bit too long. It is clearly a good point to justify why the studied parameters were selected. However, this could be summarized in the introduction and discussed more extensively in the discussion section.
  3. The sentence line 51-57 is too long. Please split it.
  4. CRP, D-dimer and cystatin C are not cytokine as it is suggested by the formulation line 54. Please clarify.
  5. The paragraph between line 59 and line 66 could be removed and combined with the last 2 paragraphs line 140-148.
  6. Please change LDL by LDL-C (LDL Cholesterol) and HDL by HDL-C (HDL Cholesterol) throughout the manuscript.
  7. The sentence “HIV-infected men were also analyzed in sub-groups (line 161-164) is not clear and at the good place. This should be described in the statistical analysis section.
  8. The paragraph line 172-178 should be placed at the beginning of the section devoted to the determination of selected parameters in plasma.
  9. Line 205-208: this important point should be placed at the beginning of the section describing the patient selection and the study design.
  10. Table 3 indicates post-hoc analysis. Its significance is not clear for the reader; this should be described in the statistical analysis section.
  11. Line 264-265 “depending on the CD4+ T cells count… CD8+ T cells count”. Please indicate that it is at the sampling time, as it is believed.
  12. Table 4: please change the header of the column “CD4+ T cells ≤1000” by “CD4+ T cells ≤300”.
  13. Table 5: please remove the section devoted to HIV-infected men after cART since there is no patient with high level of HIV RNA copies (thanks to the efficacy of therapy!). Just mention this point in the text.
  14. Legend to A, D, F and G figures of supplementary material is not correct. After cART should be dark grey.

Major points:

  1. Table 4: The interest of this comparison between high and low level of CD4 or CD8 is not clear, since a majority (35/53 for CD4 and 27/53 for CD8) presented with an acceptable level before treatment. Did the authors study the difference between those patients experiencing an improvement in CD4 or CD8 level and those keeping a non-acceptable level? Or do they think that this is not relevant?
  2. Sentence line 277-280 does not fit with what is shown on table 4. In this table, the authors did a comparison between before and after cART; they should not comment on a difference between low and high level of CD4. The same remark applies to lines 289-291.
  3. The authors should avoid comment on non-significant changes (lines 280, 287, 291, 329).
  4. The discussion line 340-343 is not correct. To say that “the levels of selected parameters… were compared… depending on the clinical status of patients…”, the authors should have done a multivariate analysis including clinical status as co-variates. This is not what is presented here. Did the authors think about such an analysis?
  5. The comment presented lines 447-449 appears a little bit too affirmative. It does not appear clearly in the presented results that “the improvement of clinical condition and immune system function can be also associated with the decrease of CD8+ T-cell count and an increased level of FETU-A”.
  6. The comment presented lines 477-478 is not correct. To conclude that “… no correlation was found between the level of CD4…” a correlation analysis should have been done. In addition, the sentence is not correct; it should be told “… no correlation was found between…. and …. (something else).
  7. The comment, lines 478-480 is not correct. To conclude to a prognostic role of SDF-1, the authors should have run a prospective study. This is obviously not the case.
  8. Again, in conclusion, the authors go back to non-significant results (lines 514-517). If they want to insist on these non-significant results, they should explain why it is important to mention it.
  9. Comment lines 515-517 is not clear at all. “No statistically significant differences…” between what and what? Moreover, “…. Depending on CD4…” did the author run a multivariate analysis?

Author Response

Answers to comments to manuscript  jcm-1616885 Reviewer 1.

Thank you for your time and checking our manuscript entitled: “Effect of combined antiretroviral therapy on the levels of selected parameters reflecting metabolic and inflammatory disturbances in HIV-infected patients” and substantive comments. We agree with all suggestions and made changes to the manuscript document using "track changes". We believe that the following changes significantly improved the quality of our manuscript.

As recommended, the manuscript was corrected according to the following points:

  1. The abstract mentions statistically significant differences without any detail (line 22). Some numbers should be given for those significant differences: median (IQ) and p value.

Response

The p values have been added for individual parameters in the indicated fragment.: “The analysis of HIV-infected patients before and after cART as well as the control group showed statistically significant differences for the following parameters: IRS (p=0.02) , MSTN (p=0.03) , PYY (p=0.03), GLP-1 (p=0.03), PTX3 (p=0.03) and RANTES (p=0.02), but no significant differences were found for DPP-4, FETU-A and SDF-1.”

Due to limitations in the number of words in the abstract, as indicated in the instructions for the authors, it is not possible to include the median and IQR values in the abstract. These values are presented in Table 3.

  1. The introduction appears a little bit too long. It is clearly a good point to justify why the studied parameters were selected. However, this could be summarized in the introduction and discussed more extensively in the discussion section.

Response

As suggested, the Introduction section has been significantly shortened. More detailed information on the parameters tested is provided in the Discussion section.

  1. The sentence line 51-57 is too long. Please split it.

Response

The sentence has been shortened and modified: “One of the causes of metabolic disorders, in addition to lifestyle and the use of cART, is the state of chronic inflammation. Increased expression of cytokines and other proinflammatory factors (e.g.: soluble CD163, CD40, CD27, Interleukin -6, CRP, D-dimer, cystatin C), especially in the area of adipose tissue, liver, skeletal muscles, and the digestive system, causes changes in metabolism and increased storage of adipose tissue, predisposing to type 2 diabetes mellitus (T2DM) and cardiovascular diseases (CVD).”

  1. CRP, D-dimer and cystatin C are not cytokine as it is suggested by the formulation line 54. Please clarify.

Response

Corrected, as suggested: “Increased expression of cytokines and other proinflammatory factors (e.g.: soluble CD163, CD40, CD27, Interleukin -6, CRP, D-dimer, cystatin C), especially in the area of adipose tissue, liver, skeletal muscles, and the digestive system, causes changes in metabolism and increased storage of adipose tissue, predisposing to type 2 diabetes mellitus (T2DM) and cardiovascular diseases (CVD)”.

  1. The paragraph between line 59 and line 66 could be removed and combined with the last 2 paragraphs line 140-148

Response

As suggested, this fragment has been moved to the end of the Introduction section and modified:” In the presented study, an attempt was made to assess changes in the level of selected parameters characterizing carbohydrate and lipid metabolism, cardiovascular diseases and inflammation (IRS, MSTN, PYY, GLP-1, DPP-4, FETU-A, PTX3, SDF-1 and RANTES) in the plasma of HIV-infected subjects before and one year after the implementation of cART.

The aim of the study was to assess the effect of cART on the level of selected parameters in the plasma of HIV-infected subjects. The analyses took into account the influence of parameters characterizing the state of the immune system, such as CD4+T cells and CD8+T cells count, HIV RNA viral load as well as the type of antiretroviral treatment regimen used: PIs or INSTIs.”

  1. Please change LDL by LDL-C (LDL Cholesterol) and HDL by HDL-C (HDL Cholesterol) throughout the manuscript.

Response

Corrected as suggested (lines 242, 243, 298, 303, 304, 308, 646)

  1. The sentence “HIV-infected men were also analysed in sub-groups (line 161-164) is not clear and at the good place. This should be described in the statistical analysis section.

Response

As suggested, moved to the Statistical analysis subsection.

  1. The paragraph line 172-178 should be placed at the beginning of the section devoted to the determination of selected parameters in plasma.

Response

As suggested, moved to the beginning of the Determination of selected parameters levels in plasma of HIV-infected men and healthy controls subsection.

  1. Line 205-208: this important point should be placed at the beginning of the section describing the patient selection and the study design.

Response

As suggested, moved to the beginning of the Materials and Methods section.

  1. Table 3 indicates post-hoc analysis. Its significance is not clear for the reader; this should be described in the statistical analysis section.

Response

As suggested, a sentence has been added: “The Kreskas-Wallis test was used to compare three groups (HIV-infected men before cART, after cART and controls) in terms of the quantitative variables studied. The statistically significant result of the Kreskas-Wallis test indicates that at least one group differs from the other group. Therefore, a post-hoc test (Dunn test with Bonferroni correction) was then performed to see exactly which groups differ from each other.”

  1. Line 264-265 “depending on the CD4+ T cells count… CD8+ T cells count”. Please indicate that it is at the sampling time, as it is believed

Response

As suggested, we added a fragment:” The values of selected parameters in HIV-infected men (both before cART (A) and after treatment (B), depending on the CD4+ T cells count (below and above 300 cells/µL) and on the CD8+ T cells count (below and above 1000 cells/µL), at the time of sampling are presented in Table 4.” (Lines 354-357)

  1. Table 4: please change the header of the column “CD4+ T cells ≤1000” by “CD4+ T cells ≤300”.

Response

As suggested, we corrected the header of the column of Table 4.

  1. Table 5: please remove the section devoted to HIV-infected men after cART since there is no patient with high level of HIV RNA copies (thanks to the efficacy of therapy!). Just mention this point in the text.

Response

As suggested, we corrected the Table 5, and added a fragment in the text:” The concentration and the results of statistical analysis of selected parameters in HIV-infected men before cART (A) depending on the amount of HIV RNA copies (below and above 100 000 copies/mL), at the time of sampling, are presented in Table 5. None of the HIV-infected men with a pre-treatment viral load HIV RNA>100 000 [copies /mL] maintained such a load after cART (B).”

  1. Legend to A, D, F and G figures of supplementary material is not correct. After cART should be dark grey

Response

Corrected, as suggested.

Major points:

  1. Table 4: The interest of this comparison between high and low level of CD4 or CD8 is not clear, since a majority (35/53 for CD4 and 27/53 for CD8) presented with an acceptable level before treatment. Did the authors study the difference between those patients experiencing an improvement in CD4 or CD8 level and those keeping a non-acceptable level? Or do they think that this is not relevant?

Response

The authors created subgroups of patients before and after cART, based on CD4 + and CD8 + T cell count, as shown in Table 4. The authors compared levels of selected parameters in  patients with high  CD4 + or CD8 + T cell count in the group before and after cART, and the same comparison was made in patients with low CD4 + and CD8 + T cell count, as suggested by the attending physician, monitoring the patients.

Statistical analysis was also performed comparing patients before cART, depending on high and low CD4+ and CD8 + T cell count and the same analysis in the post-treatment group, respectively.

In both cases, no statistically significant differences were found.

We agree with the reviewer's suggestion that an analysis comparing patients with or without improvement in CD4 + and CD8 + T cell count would be useful, but due to the low number of such patients in the study group, it is not possible to perform this type of analysis.

In our opinion, it will also be reasonable to conduct further analyses and increase the size of the studied population, which is planned in our further research.

  1. Sentence line 277-280 does not fit with what is shown on table 4. In this table, the authors did a comparison between before and after cART; they should not comment on a difference between low and high level of CD4. The same remark applies to lines 289-291.

Response:

We agree with the above remark and have been removed from the text, as suggested.

  1. The authors should avoid comment on non-significant changes (lines 280, 287, 291, 329).

Response

We agree with the above remark and comments on non-significant changes have been removed from the text

  1. The discussion line 340-343 is not correct. To say that “the levels of selected parameters… were compared… depending on the clinical status of patients…”, the authors should have done a multivariate analysis including clinical status as co-variates. This is not what is presented here. Did the authors think about such an analysis?

Response:

We agree that it would be necessary to perform a multivariate analysis including the clinical condition of patients. However, in the studied population, the majority of patients respond well to applied treatment. As shown in Table 4, a small number of patients do not achieve adequate LT CD4 +  cell count after one year of treatment and none of the HIV-infected men after cART achieved  viral load  HIV RNA>100 000 [copies /mL].  In order to perform this type of analysis, according to the authors, it is necessary to collect also a group of patients, who do not show an adequate response to treatment.

The quoted sentence was corrected: “The levels of selected parameters (IRS, MSTN, PYY, GLP-1, DPP-4, FETU-A, PTX 3, SDF-1, RANTES) were compared, both before and one year after the cART application, depending on the value of individual parameters characterizing the clinical status of patients (CD4+T cells and CD8+T cell count, HIV RNA viral load) and the type of treatment used (PIs or INSTIs).”

  1. The comment presented lines 447-449 appears a little bit too affirmative. It does not appear clearly in the presented results that “the improvement of clinical condition and immune system function can be also associated with the decrease of CD8+ T-cell count and an increased level of FETU-A”.

Response

We agree with the suggestion, the following amendments were made: “The obtained results indicate that the level of FETU-A in HIV-infected men increases as a result of the applied treatment. However, more research is needed to determine the usefulness of FETU-A in monitoring the course of HIV infection and the effectiveness of therapy.”

  1. The comment presented lines 477-478 is not correct. To conclude that “… no correlation was found between the level of CD4…” a correlation analysis should have been done. In addition, the sentence is not correct; it should be told “… no correlation was found between…. and …. (something else).

Response:

Corrected, as suggested : “In our own study, no significant differences were found between the level of SDF-1 in subgroups depending on CD4+ T cell count (≤ 300 cells/µL and >300 cells/ µL) in HIV-infected men before and after cART.”

  1. The comment, lines 478-480 is not correct. To conclude to a prognostic role of SDF-1, the authors should have run a prospective study. This is obviously not the case.

Response:

Corrected:” In our own study, no significant differences were found between the level of SDF-1 in subgroups depending on CD4+ T cell count (≤ 300 cells/µL and >300 cells/ µL) in HIV-infected men before and after cART. However, further studies are needed to confirm the prognostic role in poor immunological recovery for SDF-1 in patients with lower CD4+ T cell count.”  

  1. Again, in conclusion, the authors go back to non-significant results (lines 514-517). If they want to insist on these non-significant results, they should explain why it is important to mention it.

Response:

We agree with the above remark and we removed from the text comments on non-significant results. Non-significant results were commented both in the Results and in the Discussion section.

  1. Comment lines 515-517 is not clear at all. “No statistically significant differences…” between what and what? Moreover, “…. Depending on CD4…” did the author run a multivariate analysis?

Response:

Following the reviewer's suggestion, fragments were removed from the text.

We agree that this type of formulation was not correct.

We hope that the amendments significantly improved the quality of our manuscript and enable its publication in Journal of Clinical Medicine.

Reviewer 2 Report

In this study authors investigated changes in the level of selected parameters characterizing carbohydrate and lipid metabolism, cardiovascular diseases and inflammation in samples from HIV-infected subjects before and one year after the implementation of cART. They found significant changes in the levels of selected parameters as a result of HIV infection and changes along with the implementation of cART even in a short observation period of one year.

The study is interesting, but I have some concern about its description.

Major points

  1. Overall a more in-depth discussion to clarify the meaning of the present results would be desirable.
  2. Some comments regarding translational implications should be added in the conclusion of both abstract and manuscript.
  3. HIV strain should be specified.

Minor points

  1. Lines 14-17 please sobstitute all the semicolon with comma
  2. Lines 62-65 you should use only the acronyms
  3. Line 46 please cyte more recent articles on the topic
  4. Table 1 please align the values
  5. Please add space after tables legend
  6. Line 265 add a space
  7. Table 4 Results should be of selected parameters before and after cART in HIV-infected men by CD4+ T cells count below and above 300 cells/μL but it is written CD4+ T cells ≤1000 in the second half of the table.
  8. Line 542 link not correct
  9. Figure 1 is not clear, a legend should be added after each graph
  10. Please check punctuation and brackets in Figure 1 legend.

Author Response

Answers to comments to manuscript  jcm-1616885 Reviewer 2.

Thank you for your time and checking our manuscript entitled; ‘Effect of combined antiretroviral therapy on the levels of selected parameters reflecting metabolic and inflammatory disturbances in HIV-infected patients” and substantive comments. We agree with all suggestions and made changes to the manuscript document using "track changes". We believe that the following changes significantly improved the quality of our manuscript.

As recommended, the manuscript was corrected according to the following points:

  1. Overall a more in-depth discussion to clarify the meaning of the present results would be desirable.

The discussion was significantly modified, more information on the parameters tested was added to explain the meaning of the results and to justify the selection of individual parameters. It is the first study of this type, therefore it is not possible to indicate the exact meaning of the results obtained at this stage. However, they indicate the need for further research in this area.

  1. Some comments regarding translational implications should be added in the conclusion of both abstract and manuscript.

We added suggested information in the Abstract section: “The obtained data will allow for better monitoring of the course of HIV infection and optimization of therapy in order to prevent the development of comorbidities as a result of long-term use of cART.” and in the Conclusion section: “The obtained data may be useful in monitoring the course of HIV infection, treatment efficacy or optimization of therapy. This may be helpful in the prevention or development of comorbidities associated with HIV infection and chronic cART use.”

HIV strain should be specified.

All patients were infected with the HIV-1 strain. We added information in Material and Methods section. 

Minor points

  1. Lines 14-17 please sobstitute all the semicolon with comma

Corrected

  1. Lines 62-65 you should use only the acronyms

Corrected

  1. Line 46 please cyte more recent articles on the topic

We added some references: “There are different therapeutic regimens based on the application of at least two nucleoside reverse transcriptase inhibitors (NRTIs) in combination with non-nucleoside reverse transcriptase inhibitors (NNRTIs), integrase transfer inhibitors (INSTIs), protease inhibitors (PIs), fusion inhibitors and C-C Chemokine Receptor 5 (CCR5) antagonists. In addition to two NRTIs, recommended regimens include protease inhibitors (PIs) or integrase transfer inhibitors (INSTIs) [3-5].”

  1. Pau, A.K.; George, J.M. Antiretroviral therapy: Current drugs. Infect. Dis. Clin. North Am. 2014, 28, 371–402.
  2. Ryom, L.; Cotter, A.; De Miguel, R.; Béguelin, C.; Podlekareva, D.; Arribas, J.R.; Marzolini, C.; Mallon, P.G.M.; Rauch, A.; Kirk, O.; et al. 2019 update of the European AIDS Clinical Society Guidelines for treatment of people living with HIV ver-sion 10.0. HIV Med. 2020, 21, 617–624, doi:10.1111/hiv.12878.
  3. Consolidated Guidelines on HIV Prevention, Testing, Treatment, Service Delivery and Monitoring: Recommendations for a Public Health Approach. Geneva: World Health Organization; 2021 Jul. 7. Available online: https://www.ncbi.nlm.nih.gov/books/NBK572734/ (accessed on 08 March 2022).
  4. Table 1 please align the values

Corrected

  1. Please add space after tables legend

Corrected

  1. Line 265 add a space

Corrected

  1. Table 4 Results should be of selected parameters before and after cART in HIV-infected men by CD4+ T cells count below and above 300 cells/μL but it is written CD4+ T cells ≤1000 in the second half of the table.

Corrected

  1. Line 542 link not correct

Corrected

  1. Figure 1 is not clear, a legend should be added after each graph

We added a legend after each graph (supplementary material).

  1. Please check punctuation and brackets in Figure 1 legend.

Corrected. We added a legend after each graph (supplementary material).

We hope that the amendments significantly improved the quality of our manuscript and enable its publication in Journal of Clinical Medicine.

Round 2

Reviewer 1 Report

Thanks to the authors for this new version. They answered correctly to all remarks. I have no further request. 

Author Response

Thank you for your feedback.

We hope that the amendments significantly improved the quality of our manuscript and enable its publication in Journal of Clinical Medicine.

Thank you very much for your time.

Best regards